# Aging-regulated TUG1 is dispensable for endothelial cell function

**Anna Theresa Gimbel**[1,2]**, Susanne Koziarek**[1]**, Kosta Theodorou**[1]**, Jana Felicitas Schulz**[3,4]**, Laura Stanicek**[1,5]**, Veerle Kremer**[5]**, Tamer Ali**[1]**, Stefan Günther**[2,6]**, Sandeep Kumar**[7,8]**, Hanjoong Jo**[7,8]**, Norbert Hübner**[3,4,9]**, Lars Maegdefessel**[10,11,12]**, Stefanie Dimmeler**[1,2]**, Sebastiaan van Heesch**[13]**, Reinier A. Boon**[1,2,5]*

1 Centre of Molecular Medicine, Institute of Cardiovascular Regeneration, Goethe-University, Frankfurt am Main, Germany, 2 DZHK (German Centre for Cardiovascular Research), Partner Site Rhine-Main, Frankfurt, Germany, 3 Cardiovascular and Metabolic Sciences, Max Delbrück Center for Molecular Medicine in the Helmholtz Association (MDC), Berlin, Germany, 4 DZHK (German Centre for Cardiovascular Research), Partner Site Berlin, Berlin, Germany, 5 Department of Physiology, Amsterdam Cardiovascular Sciences, VU University Medical Centre, Amsterdam, Netherlands, 6 Max Planck Institute for Heart and Lung Research, Bioinformatics and Deep Sequencing Platform, Bad Nauheim, Germany, 7 Wallace H. Coulter Department of Biomedical Engineering, Georgia Institute of Technology and Emory University, Atlanta, Georgia, United States of America, 8 Division of Cardiology, Emory University, Atlanta, Georgia, United States of America, 9 Charité-Universitätsmedizin, Berlin, Germany, 10 Department of Vascular and Endovascular Surgery, Technical University Munich, Munich, Germany, 11 German Center for Cardiovascular Research DZHK, Partner Site Munich, Munich, Germany, 12 Department of Medicine, Karolinska Institute, Stockholm, Sweden, 13 Princess Máxima Center for Pediatric Oncology, Utrecht, The Netherlands

* r.a.boon@amsterdamumc.nl

**Data Availability Statement:** The RNA sequencing data for this study is deposited at Gene Expression Omnibus with accession number: GSE200766

## Abstract

The evolutionary conserved *Taurine Upregulated Gene 1* (*TUG1*) is a ubiquitously expressed gene that is one of the highest expressed genes in human and rodent endothelial cells (ECs). We here show that *TUG1* expression decreases significantly in aging mouse carotid artery ECs and human ECs *in vitro*, indicating a potential role in the aging endothelial vasculature system. We therefore investigated if, and how, *TUG1* might function in aging ECs, but despite extensive phenotyping found no alterations in basal EC proliferation, apoptosis, barrier function, migration, mitochondrial function, or monocyte adhesion upon *TUG1* silencing *in vitro*. *TUG1* knockdown did slightly and significantly decrease cumulative sprout length upon vascular endothelial growth factor A stimulation in human umbilical vein endothelial cells (HUVECs), though TUG1-silenced HUVECs displayed no transcriptome-wide mRNA expression changes explaining this effect. Further, ectopic expression of the highly conserved and recently discovered 153 amino acid protein translated from certain *TUG1* transcript isoforms did not alter angiogenic sprouting *in vitro*. Our data show that, despite a high expression and strong evolutionary conservation of both the *TUG1* locus and the protein sequence it encodes, *TUG1* does not seem to play a major role in basic endothelial cell function.

(https://www.ncbi.nlm.nih.gov/geo/query/acc.cgi?acc=GSE200766).

**Funding:** RB was supported by the Rembrandt Institute of Cardiovascular Science, the German Centre for Cardiovascular Research (DZHK), the European Research Council (ERC starting grant "NOVA" and consolidator grant "NICCA"), the European Union (Horizon 2020 Grant No. 825670), the Cardiopulmonary Institute (CPI) and the SFB834 and TRR267 of the German Research Council (DFG). NH was supported by the ERC Advanced Grant "CodingHeart". The funders had no role in study design, data collection and analysis, decision to publish, or preparation of the manuscript.

**Competing interests:** The authors have declared that no competing interest exist

## Introduction

In the last decade, non-coding RNAs, and especially long non-coding RNAs (lncRNAs), were implicated in the development of aging-induced cardiovascular diseases (CVDs) [1–4]. *Taurine upregulated gene 1 (TUG1)*, a well-studied lncRNA in many types of cancer [5], was previously associated with diabetic retinopathy in mice being an aging-induced disease [6–8]. Furthermore, *TUG1* has been described to be involved in tumor-induced angiogenesis [9, 10]. However, the role of *TUG1* in aging-induced CVDs remains largely unknown.

CVDs are the leading cause of death worldwide [11]. Each year CVDs cause over 1.8 million deaths in the European Union including myocardial infarction, stroke, atrial fibrillation, vascular diseases and many more [12]. A prominent risk factor for the development of CVDs is aging. The increasing number of the elderly is already a great challenge for the health care system that needs to be engaged within the next years. On top of that, the population over 65 years is estimated to double from 12% in 2010 to 22% in 2040 [13].

At structural level, aging correlates with several changes in the vasculature: It leads to stiffening of the vessel wall, thickening of the intima, endothelial dysfunction and increased vascular inflammation [14]. Additionally, aging leads to impaired angiogenesis, and a diminished angiogenic response to injuries, both important mechanisms in the development of CVDs [15]. Angiogenesis describes the outgrowth of new vessels from pre-existing ones via a cascade of highly coordinated cellular functions driven by pro-angiogenic stimuli [16]. Embryonic development, patterning of the vascular system and wound healing rely on the precise coordination of migrating and quiescent endothelial cells [17]. Conversely, pathological angiogenesis is involved in malignant, inflammatory, immune and ischemic disorders [16, 18].

LncRNAs are commonly characterized to be more than 200 nucleotides long, poorly conserved among species and expressed in a tissue-specific manner [19]. Via control of epigenetic [20, 21], transcriptional [22, 23] and post-transcriptional processes [24, 25], lncRNAs regulate various biological functions. *TUG1* is, in contrast to other well-studied lncRNAs, highly conserved among different species and ubiquitously expressed with moderate to high expression in different adult tissues in human and mouse [26, 27].

*TUG1* was initially identified as a crucial lncRNA in the development of photoreceptors in the mouse retina [28]. In cancer, *TUG1* acts in a tissue- or context-specific manner either as tumor-suppressor or oncogene by affecting cancer cell proliferation, migration and invasion [5, 21, 27–30]. Mechanistically, *TUG1* can recruit Polycomb-repressive complex 2 (PRC2) and repress the expression of specific target genes in *trans* in the nucleus [30]. Furthermore, *TUG1* can sequester micro RNAs (miR) in the cytoplasm, e.g. miR-145 in gliomas, further implementing a role in (epi-) transcriptomic regulation [21].

The general expectation that non-coding RNAs do not exhibit any coding potential is currently confronted. Emerging bioinformatics data and large-scale transcriptomic analyses propose the translation of a larger portion of the genome than previously accepted [31]. Instead, it is believed that about 22% of the transcribed lncRNAs are translated into microproteins [26]. Representative lncRNAs that were identified to show translation, are *Long Intergenic Non-Protein Coding RNA, P53 Induced Transcript (LINC-PINT)*, *Differentiation Antagonizing Non-Protein Coding RNA (DANCR)*, *Plasmacytoma Variant Translocation 1 (PVT1)* and many more [26]. In this study, van Heesch et al. identified an open reading frame (ORF) in a previously misannotated 5'-leader sequence of the *TUG1* transcript starting with the non-canonical start-codon CUG. Translation of the TUG1 protein (153 amino acids) was demonstrated via sequence conservation analyses, ribosome profiling, coupled *in vitro* transcription: translation assays, and ectopic expression of tagged constructs followed by Western blot and immunofluorescence microscopy. Functionally, the TUG1 protein has been described to localize to

mitochondria and influence mitochondrial bioenergetics [26, 27]. These various lines of coding-sequence evidence, together with the high amino acid sequence conservation of the TUG1 ORF across species, over the course of this project led to *TUG1*'s official classification as a protein-coding gene (Ensembl release v100; April 2020).

The influence of *TUG1* on mitochondrial bioenergetics is further emphasized in the context of diabetic retinopathy. Murine Tug1 positively regulates Ppargc1α gene transcription and its target genes in podocytes in mice by acting as a scaffold between an enhancer element, PGC-1α and the PGC-1α promoter. The interaction between Tug1 and PGC-1α increased mitochondrial content, mitochondrial respiration and cellular ATP levels and reduced mitochondrial ROS [6]. Male Tug1 knockout mice are sterile with underlying defects including a low number of sperm and an abnormal sperm morphology originating from impaired spermatogenesis [27].

Here, we characterized the lncRNA *TUG1* and its role in endothelial cell function. We showed that *TUG1* is regulated by aging in endothelial cells *in vitro* and *in vivo*. *TUG1* silencing did not change basal endothelial function addressing proliferation, apoptosis, migration, barrier function, mitochondrial function or monocyte adhesion. We identified a small impact of *TUG1* silencing on VEGFA-stimulated angiogenic sprouting *in vitro*. *TUG1* does not influence transcription supporting a dispensability in ECs. TUG1 proteins were expressed without having any effects on VEGFA-stimulated sprouting.

## Methods

### Cell culture

HUVECs (pooled donor; Lonza) were cultured in endothelial basal medium (EBM; Lonza) supplemented with 10% fetal calf serum (FCS; Invitrogen) and EGM-SingleQuots (Lonza). Cells were cultured at 37˚C with 5% $CO_2$. For the different assays passages 2 or 3 of HUVECs were used. Human embryonic kidney cells (Hek293T; DSMZ; # ACC 305) [32] were cultured in DMEM with 10% heat-inactivated FCS, D-glucose, pyruvate and Penicillin/streptomycin. THP-1 cells (DSMZ; # ACC 16) [32] were cultured in RPMI with 10% heat-inactivated FCS and Penicillin/streptomycin. Cells were cultured at 37˚C and 5% $CO_2$. Cell numbers were determined with the Nucleocounter NC-2000 (Chemometec A/S).

### Transfection and lentiviral overexpression

HUVECs were transfected at 60% confluency with 10 nmol/L locked nucleic acid (LNA) GapmeRs (Qiagen) or small interfering RNAs (siRNAs; Sigma Aldrich) using Lipofectamine RNAiMAX (Life Technologies) according to the manufacturer´s protocol. A siRNA against firefly luciferase (Sigma Aldrich) or scrambled LNA GapmeR (Qiagen, 339516) were transfected as controls. The medium was changed to EBM (Lonza) supplemented with 10% FCS (Invitrogen) and EGM SingleQuots (Lonza) after 4h. GapmeR and siRNA sequences are listed in the Supplement.

Lentivirus stocks were produced in Hek293T cells using pCMVΔR8.91 as packaging plasmid and pMD2.G (Addgene #12259) as vesicular stomatitis virus G glycoprotein (VSV-G) envelope expressing plasmid [33]. In brief, 10x10^6 Hek293T were seeded 24h before transfection. Cells were transfected with 10 μg pLenti4-V5 plasmid, 6 μg packaging plasmid pCMVΔR8.91 and 2 μg VSV-G plasmid pMD2.G. Empty pLenti4-V5 vectors were used as control (mock). HUVECs were transduced for 24 h. Medium was changed daily until day 3 after transfection and the supernatant was combined from day 2 and day 3. Virus was concentrated by usage of LentiX Concentrator (TakaraBio) according to the manufacturer´s

instructions. The virus pellet was resuspended in 1 mL PBS (Gibco). For long term storage, virus suspension was aliquoted in 1.5 ml cryotubes with 500 μl per condition and stored at -80˚C.

One day before transfection, HUVECs were seeded in a density of $3x10^5$ cells. HUVECs were transduced with one aliquot of virus. Cells were washed 24 h and 72 h after transduction 3 times in an alternating order with PBS and EBM.

## Plasmid cloning

For the overexpression of the TUG1 ORF, three different constructs with a C-terminal FLAG-tag were subcloned from pEF1a vectors [26] into pLenti4-V5 backbones: pLenti4-V5_h-sTUG1_lncRNAshort, pLenti4-V5_hsTUG1_CTGmut and pLenti4-V5_hsTUG1_CDS. The initial step involves a PCR with the ORFs (including desired up- and downstream sequences) as a template and the addition of the TOPO site in parallel. The pENTR™/D-TOPO® Cloning Kit (Invitrogen) utilizes a highly efficient "TOPO Cloning" strategy to directionally clone the blunt-end PCR product into the pENTR vector as an entry point into the Gateway® System. Identification of positive transformants was achieved by transformation of One Shot™ Stbl3™ Chemically Competent *E. coli* (Invitrogen) and subsequent analysis by restriction analysis with EcoRV-HF (New England Biolabs) and sequencing. Gateway® LR Clonase® II enzyme mix (Invitrogen) catalyzed the *in vitro* recombination between and the entry clone pENTR (containing the gene of interest flanked by *att*L sites) and the pLenti4-V5 destination vector (containing *att*R sites) to generate an expression clone by homologous recombination. The analysis of the resulting clones was performed by restriction analysis with NcoI-HF (New England Biolabs), for the exclusion of unwanted recombination events, and sequencing.

## Western blot

HUVECs were lysed with Radioimmunoprecipitaion assay (RIPA) buffer (Thermo Fisher) supplemented with Halt™ Protease and Phosphatase Inhibitor Cocktail (1:100; Thermo Fisher). Cell lysis was enabled by an incubation at 4˚C at a turning wheel for 1 h. Cell debris was removed by centrifugation (16000 xg for 10 min at 4˚C). Protein concentration was determined by Pierce™ BCA Protein Assay Kit (Thermo Fisher Scientific). Equal amounts of denaturated protein in Laemmli buffer were loaded on 12% Sodium dodecyl sulfate gels (BioRad) and blotted on nitrocellulose membranes (Invitrogen). Membranes were blocked with 3% milk (Roth) in TBS-T and incubated with primary antibody overnight at 4˚C under rotation. Secondary antibodies tagged with Horse Radish Peroxidase (HRP; Dako) were incubated for 1 h at room temperature under rotation. Bands were visualized using enhanced chemiluminescence (ECL, Thermo Fisher) on the ChemiDoc device (BioRad). Band intensity was quantified using ImageLab Software (version 5.2.1; BioRad). Antibodies and dilutions can be found in the S1 Table and uncropped images of Western blots are available in S1 Raw image.

## RT-qPCR

Total RNA was isolated from HUVECs with the RNA Direct-zol RNA miniprep Kit (Zymoresearch) by following the manufacturer´s protocol including DNase digest. For Quantitative Real Time PCR (RT-qPCR) 100–1000 ng RNA were reverse transcribed using random hexamer primers (Thermo Fisher) and Multiscribe reverse transcriptase (Applied Biosystems). The resulting copy DNA (cDNA) was used as template for RT-qPCR in combination with Fast SYBR Green Master Mix (Applied Biosystems) in an Applied Biosystems StepOnePlus machine (Applied Biosystems) or Viia7 device (Applied Biosystems). Human ribosomal protein (RPLP0), glyceraldehyde-3-phosphate (GAPDH) or TATA-Box Binding Protein (TBP)

were used for normalization. Gene expression analysis was performed by using the $2^{-\Delta\Delta CT}$ method. Primer sequences are listed in the S1 Table.

## RNA sequencing

HUVECs were transfected with Control or TUG1 GapmeRs. Total RNA was isolated after 48 h with QIazol (Qiagen) and the Direct-zol RNA miniprep kit (Zymo Research) according to the manufacturer's instructions including DNase digest. Alternatively, total RNA was isolated from cell pellets from cardiomyocytes, aortic fibroblasts, pericytes, aortic smooth muscle cells, mesenchymal stem cells, dermal lymphatic endothelial cells, umbilical vein endothelial cells, saphenous vein endothelial cells, pulmonary microvascular endothelial cells, dermal microvascular endothelial cells, cardiac microvascular endothelial cells, coronary artery endothelial cells, pulmonary artery endothelial cells and aortic endothelial cells (all human; Promocell) with the miRNeasy Micro Kit (Qiagen) according to the manufacturer´s instructions including DNase digest. Quality control of total RNA and library integrity was assessed by LabChip Gx Touch 24 (Perkin Elmer). The library was generated by using the SMARTer Stranded Total RNA Sample Prep Kit—HI Mammalian (Clontech) with 1 μg RNA as input. Sequencing with NextSeq500 (Illumina) included v2 chemistry and 1x75bp single end setup and the derived values were analyzed for quality, adapter content and duplication rates with FastQC (Available online at: http://www.bioinformatics.babraham.ac.uk/projects/fastqc). Trimming of reads was achieved by employing Trimmomatic version 0.39 after a quality drop below a mean of Q20 in a window of 10 nucleotides [34]. Reads of at least 15 nucleotides were approved for subsequent analyses and aligned against the Ensembl human genome version hg38 (GRCh38) using STAR 2.6.1d with the parameter "—outFilterMismatchNoverLmax 0.1" to enhance the maximum ratio of mismatches to mapped length to 10% [35]. The number of reads aligning to genes was counted with featureCounts 1.6.5 from the Subread package [36]. Reads overlapping multiple genes or aligning to several genes were excluded, while reads mapping–at least partially–inside exons were accepted and collected per gene. We obtained ~35 million 75 bp single-end reads. Read mapping was done with STAR aligner using default settings with the option—outSAMtype BAM SortedByCoordinate [35] with default settings. For known transcript models we used GRCh38.R21 annotations downloaded from Gencode repository [37]. Counting reads over gene model was carried out using GenomicFeatures Bioconductor package [38]. All transcripts with read counts less than 10 were excluded. For normalization of read counts and identification of differentially expressed genes we used DESeq2 with padj < 0.05 cutoff [39].

RNA expression levels in partial carotid ligation operated age-matched male C57BL/6 mice was performed by the laboratory of H.J using 10-week-old and 18-month-old male C57Bl/6 mice (The Jackson Laboratory).

## Subcellular fractionation

Nuclear (nucleoplasm and chromatin) and cytoplasmic fractions were isolated from untransfected HUVECs. After washing with cold PBS, cells were lysed with cytoplasmic lysis buffer (10 mM Tris (pH 7.5), 150 mM NaCl, 0.15% NP-40), layered on a sucrose buffer (10 mM Tris (pH 7.5), 150 mM NaCl, 24% (w/v) sucrose) and centrifuged at 4˚C for 10 min at 16.000 x g. The supernatant (cytoplasmic fraction) was intercepted in TRIzol LS (Thermo Fisher Scientific) for RNA isolation and the pellet was resuspended in glycerol buffer (20 mM Tris (pH 7.9), 75 mM NaCl, 0.5 mM EDTA and 0.85 mM DTT, 50% glycerol). Nuclei lysis buffer (10 mM HEPES (pH 7.6), 7.5 mM MgCl2, 0.2 mM EDTA, 0.3 M NaCl, 1 M urea, 1 mM DTT, 1% NP-40) was added, incubated on ice and centrifuged for 2 min at 4˚C and 16.000 x g. The supernatant (nucleoplasm fraction) was resuspended in TRIzol LS. The pellet (chromatin

fraction) was resuspended in cold PBS and vigorously vortexed for several seconds to release the RNA. TRIzol LS was added and RNA was isolated using the Direct-zol RNA miniprep kit (Zymo Research). Equal volumes were used for subsequent reverse transcription to ensure comparison of equal cell equivalents.

### Growth curve

HUVECs were cultured in 24-well plates (Greiner) and transfected for 24 hours. The cell numbers were counted 0, 24, 48 and 72 h after transfection. For each time point, cells were washed, trypsinized (100 μl; Gibco), resuspended in PBS (300 μl) and transferred to 1.5 mL reaction tubes (Eppendorf). The tubes were vortexed and 12 μl cell suspension were transferred into Neubauer improved disposable counting chambers (NanoEntek). Cells were counted in 5 large squares (4 corner and 1 middle square). The 1.5 mL reaction tubes were spun down and the remaining volume in the 1.5 mL reaction tubes was determined. Cell number per mL and total cell number were calculated.

### ECIS and migration assay

Electrical Cell-Substrate Impedance Sensing System (ECIS; Applied BioPhysics) was used to determine the integrity of the endothelial cell barrier, as well as the potential to recover after wounding. As previously described [40], barrier-function was measured by application of an alternating current of 400 Hz. The resulting potential was detected by the ECIS instrument Zθ (Applied BioPhysics). The impedance (Ω) is calculated from the corresponding changes in voltage between electrodes according to Ohm´s law [40].

Migration analysis was based on the wounding of an intact monolayer by lethal electroporation. The recovery of endothelial cells to form a monolayer was quantified by measuring the impedance at 4000 Hz (area under the impedance curve).

### Caspase 3/7 activity assay

HUVECs were transfected 48 h before the assay. Cells were transferred to black-walled 96-well plates (Falcon) 4 h before the assay and incubated with EBM or 200nM Staurosporine (Sigma Aldrich) for 4 h. Caspase 3/7 activity was assayed according to the manufacturer´s protocol for ApoOne® Homogenous Caspase 3/7 Assay (Promega). Fluorescence was measured with Glomax Multi plate reader (Promega).

### Seahorse mitochondrial stress test

HUVECs were transfected 48 h before the assay and transferred to gelatin-fibronectin-coated specialized 96-well plates (Agilent) 24 h before the assay. Assay medium was prepared based on Seahorse XF Base Medium (Agilent) and supplemented with L-glutamine (Sigma Aldrich), glucose (Sigma Aldrich) and sodium pyruvate (Sigma Aldrich). The following protocol was performed according to the manufacturer´s instructions including calibration of the Seahorse device, serial injection of Oligomycin, FCCP, Rotenone, Antimycin A and the respective measurement of oxygen consumption rate (OCR) and extracellular acidification rate (ECAR) with Seahorse XFe96 Analyzer (Agilent). For normalization, cells were stained with Hoechst, washed with PBS and luminescence was measured with an ELISA reader. Multiple parameters including basal respiration, ATP-linked respiration and spare respiratory capacity (SRC) were collected in this assay and calculated.

## Static monocyte adhesion assay

HUVECs were transfected 48 h before the assay and transferred to gelatin-coated black-walled 96-well plates (Falcon) and stimulated with TNF-α (10 ng/ml; Peprotech) or PBS (Gibco) for 24 h. Directly before start of the assay, THP-1 cells [32] were stained with Hoechst dye (Thermo Fisher) and added to the HUVECs after stringent washing. Baseline fluorescence was measured in an ELISA reader (Promega) following an incubation of 30 min at 37˚C, 5% $CO_2$. Cells were washed with PBS and fluorescence was measured again.

## Spheroid assay

Endothelial angiogenesis was studied by spheroid sprouting assay *in vitro*. HUVECs were transfected with siRNAs or LNA GapmeRs for 24 h. Cells were trypsinized and resuspended in a mixture of culture medium and 0.6 gr/L methylcellulose (Sigma) in a ratio of 80%:20%. Cells were seeded (400 cells per 100 μl) in a U-bottom-shaped 96-well plate (Greiner) to allow the formation of spheroids for 24 h at 37˚C. The spheroids were collected, added to methylcellulose (2.4 gr/L) with FBS in a ratio of 80%:20% (Gibco) and embedded in a collagen type I (Corning) gel containing 3.77 g/L collagen I (Corning, USA), 10% M199 medium (Sigma Aldrich), 0.018 M HEPES (Invitrogen) and 0.2 M NaOH to adjust pH to 7.4. The mixture with the spheroids was allowed to polymerize for 30 minutes in a 24 well plate. Following incubation for 24 h at 37˚C with or without VEGFA (50 ng/ml; Peprotech) the gels were fixed with 10% formaldehyde (Roth) and microscope images were taken at 10x magnification (AxioVert microscope, Zeiss). The cumulative length of sprouts was quantified using the image analysis software ImageJ.

## Statistical analysis

Data are represented by mean ± standard error of mean (SEM). GraphPad Prism 7 and 9 were used for statistical analysis. Gaussian distribution was tested using Shapiro-Wilk test. Paired or unpaired Student´s t-test or Mann-Whitney tests were performed when comparing two groups. For the comparison of more than two groups Analysis of variance (ANOVA) was applied. Significant outliers within a group ($p < 0.05$) were detected by Grubbs' outlier test and excluded from the analysis. Data were considered statistically significant below a p-value of 0.05. The sample size n states the number of independent experiments.

## Results

### *TUG1* is one of the highest expressed lncRNAs in human and mouse in endothelial cells and its expression decreases strongly during aging

With the hypothesis that the highest expressed lncRNAs may govern key processes in ECs, we performed RNA-sequencing to identify novel players in EC biology. Among the top 10 was *TUG1* next to well characterized lncRNAs *Metastasis Associated Lung Adenocarcinoma Transcript 1 (MALAT1)*, *Nuclear Paraspeckle Assembly Transcript 1* (*NEAT1*) and *Maternally Expressed 3 (MEG3*; Fig 1A). These lncRNAs showed comparable expression levels as to the EC-specific protein-coding gene *Vascular Endothelial Growth Factor Receptor 2* (*VEGFR2; KDR*). As previously reported, *TUG1* is ubiquitously expressed in many organs in human and mouse [27]. We therefore analyzed *TUG1* expression in various cell types of the cardiovascular system including cardiomyocytes, fibroblasts, mesenchymal stem cells, smooth muscle cells and pericytes (Fig 1B). Endothelial cells were further divided into endothelial subtypes derived from different origins such as dermal, pulmonary and cardiac microvasculature, saphenous vein and aorta by virtue of their strong heterogeneity. These results showed that TUG1 is

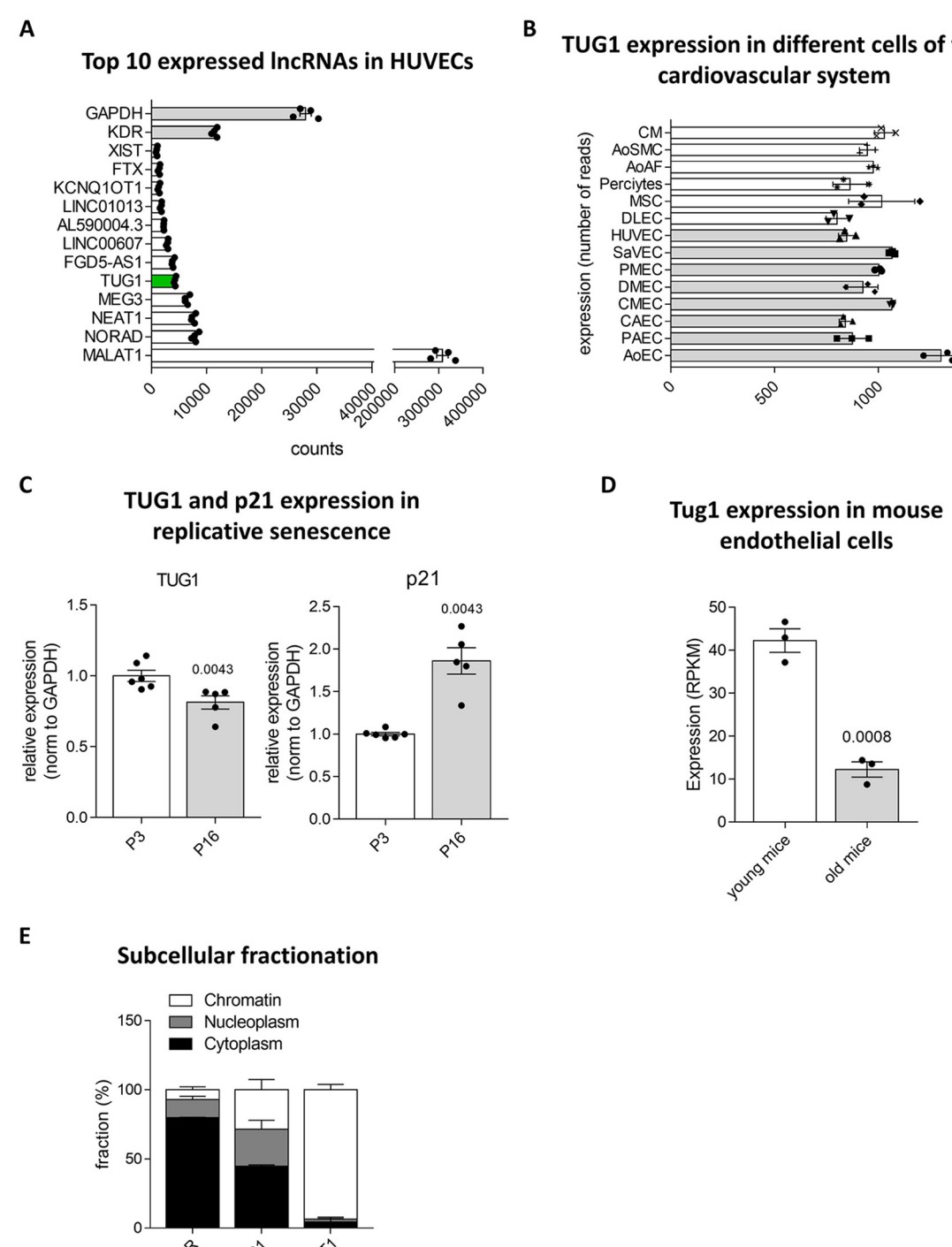

**Fig 1. *TUG1* is highly expressed in endothelial cells and regulated by aging in human and mouse. (A)** Top 10 expressed lncRNAs based on transcript counts from HUVEC bulk RNA sequencing data (n = 4). *TUG1* is highlighted in green. *Glyceraldehyde 3-phosphate dehydrogenase* (*GAPDH*) and *Kinase Insert Domain Receptor* (*KDR*) were used as controls. **(B)** RNA expression levels of *TUG1* in different human cell types of the cardiovascular system (n = 3). Vascular ECs are highlighted by grey bars. AoEC: Aortic ECs, PAEC: Pulmonary Artery ECs, CAEC: Coronary Artery ECs, CMEC: Cardiac Microvascular ECs, DMEC: Dermal Microvascular ECs, PMVEC: Pulmonary Microvascular ECs, SaVEC: Saphenous Vein ECs, HUVEC: Human Umbilical Vein ECs, DLEC: Dermal Lymphatic ECs, MSC: Mesenchymal Stem Cells, AoAF: Aortic Arterial Fibroblasts, AoSMC: Aortic Smooth Muscle Cells, CM: Cardiomyocytes **(C)** *TUG1* expression levels in low (P3) vs. high (P16) passage HUVECs as determined by RT-qPCR. Expression is relative to *GAPDH* (n = 5–6; SEM; Mann-Whitney-test). **(D)** *Tug1* expression from bulk RNA-

sequencing data of the intima of the carotid arteries of young (10 weeks) vs. aged mice (18 months) (n = 3; SEM; Mann-Whitney-test). **(E)** Quantification of the expression levels of the lncRNAs *Differentiation Antagonizing Non-Protein Coding RNA* (*DANCR*), *TUG1* and *Metastasis Associated Lung Adenocarcinoma Transcript 1* (*MALAT1*) in subcellular fractions of wild type HUVECs using RT-qPCR (n = 3). Results are expressed as percentages of the subcellular fractions associated to cytoplasm, nucleoplasm and chromatin.

ubiquitously expressed across cell types analyzed. *TUG1* RNA levels were further analyzed in the context of replicative senescence in human ECs *in vitro* (Fig 1C), to assess a potential role of *TUG1* during aging of the vascular system. *TUG1* levels were slightly, but significantly decreased in high passage HUVECs compared to HUVECs in passage 3, whereas *p21* was induced in passage 16 HUVECs as expected. Complementary, *Tug1* expression was attenuated in aged (18 months) compared to young mice (10 weeks), to a high extent (Fig 1D). Taken together, *TUG1* is highly expressed in endothelial cells and downregulated upon aging in human and mouse. RNA molecules can feature various functions within the cell depending on their subcellular localization [41]. Therefore, nuclear (separated into nucleoplasm and chromatin) and cytoplasmic fractions were isolated and analyzed by RT-qPCR (Fig 1E). *Differentiation antagonizing non-protein coding RNA* (*DANCR*) and (*MALAT1*) served as controls. *MALAT1* is exclusively associated to chromatin [42], whereas *DANCR* is a well characterized transcript known to be mainly cytoplasmic [43], where it is also translated [26]. Consistent with previous results obtained within other cell types [44–46], *TUG1* was equally distributed across the nucleoplasm, the chromatin, and cytoplasm within ECs, suggesting that *TUG1* might incorporate different functions in ECs.

### *TUG1* is not involved in proliferation, apoptosis, migration, barrier function, mitochondrial function, and inflammation under basal conditions in ECs

To simulate the reduced *TUG1* levels in aged human ECs and investigate a potential role of *TUG1*, HUVECs were transfected with Locked Nucleic Acid (LNA) GapmeRs to reduce the high abundance of the *TUG1* transcript in low passage HUVECs. LNA GapmeRs are short single-stranded DNA oligonucleotides that are flanked by LNA nucleotides. Total *TUG1* levels were strongly decreased by two different LNA GapmeRs (LNA TUG1_1–14.46% ± 3.28%; LNA TUG1_2–16.40% ± 3.37%) compared to LNA Ctrl (100% ± 5.13%) using RT-qPCR (Fig 2A). Following this, effects of *TUG1* silencing on EC function were determined. An important hallmark of aging is the reduction of cell proliferation and increased inflammation [14]. However, cell turnover–including cell count (Fig 2B) and apoptosis (Fig 2C)–were not changed by loss of *TUG1* compared to control. Another EC-specific characteristic addresses barrier function. The method Electric Cell Impedance Sensing (ECIS) analyzes the morphology, which allows to study cell-cell or cell-matrix interactions can be studied. As Fig 2D displays, *TUG1* silencing had no effect on either of these interactions. ECIS was also used to determine the migratory capability. For this purpose, a high frequency current was applied and a cell-free area was created by electroporation. Cells from the surrounding area migrate to re-establish a monolayer which can be determined by the change in impedance. The slope of the curve was similar in the control and both *TUG1* knockdown conditions, indicating that cells migrated at a similar speed (Fig 2E).

To assess the role of Tug1 in cellular metabolism, parameters assigned to mitochondrial stress (basal respiration, maximal respiration, proton leak, ATP production and spare respiratory capacity) were analyzed after loss of *TUG1* using the Seahorse platform. None of the mitochondrial stress characteristics were influenced by loss of *TUG1* (Fig 2F). A further

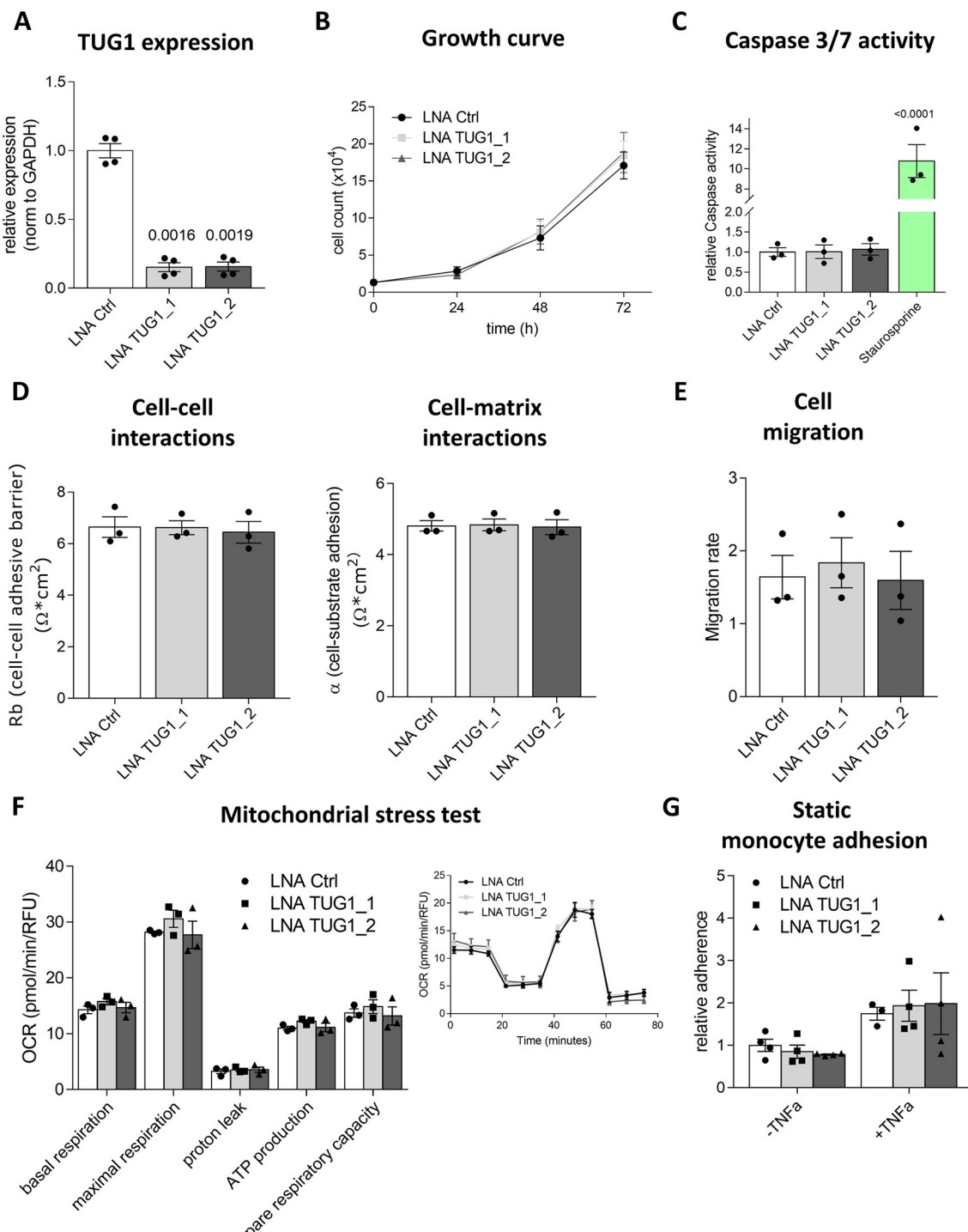

**Fig 2. TUG1 is not important for basal cell turnover, barrier or mitochondrial function, migration and monocyte adhesion. (A)–(G)** HUVECs were transfected with two LNA GapmeRs against *TUG1*—LNA TUG1_1 and LNA TUG1_2 –and LNA Ctrl (10 nM) and **(A)** expression levels were measured after 48 hours by RT-qPCR. Expression is relative to *GAPDH* (n = 4; SEM; RM one-way ANOVA with Greenhouse-Geisser correction and Sidak multiple comparison test). **(B)** Relative cell growth determined from cell count at 0 h, 24 h, 48 h and 72 h (n = 3; SEM; RM Two-way ANOVA with Tuckey multiple comparison test). **(C)** Caspase-3/7 activity was measured by determination of fluorescence with ELISA plate reader (n = 3; SEM; One-way ANOVA with Holm-Sidak correction). Staurosporine was taken along as a postive control. **(D)** Cell-cell interactions (Rb) and cell-matrix-interactions (α) were measured by Electric Cell Impedance Sensing (ECIS; n = 3; SEM; Kruskal-Wallis-test with Dunn´s correction). **(E)** Determination of re-establishment of monolayer after wounding using ECIS (n = 3; SEM; One-way ANOVA with Holm-Sidak multiple comparison test).

(F) Seahorse mitochondrial stress test assessing multiple mitochondrial characteristics via measurement of changes in Oxygen Consumption Rate (OCR) after serial injection of Oligomycin, Carbonyl cyanide-4 (trifluoromethoxy) phenylhydrazone (FCCP) and Rotenone A/Antimycin (n = 3; SEM; One-way ANOVA with Holm-Sidak multiple comparison test. One representative experiment displaying the changes of OCR throughout the progress of the Seahorse mitochondrial stress test assay. (G) Assessment of monocyte adhesion with and without TNF-α stimulation. (n = 3; SEM; Two-way ANOVA with Tuckey multiple comparison test).

characteristic of cardiovascular aging is a low-grade chronic inflammation [47, 48]. To target this feature, *TUG1* was knocked down in HUVECs and the effect on adhesion of monocytes was assessed (Fig 2G). *TUG1* manipulation had no effect on monocyte adhesion in untreated and TNF-α stimulated HUVECs. Stimulation with TNF-α was used as a positive control as monocyte adhesion is increased. We tested different chemical stimuli such as oxLDL, $H_2O_2$, VEGFA, TNF-α and Delta-like protein 4 (Dll4) and shear stress as mechanical stimulation. *TUG1* expression was not changed in response to any of the mentioned stress conditions (S1 Fig). In summary, *TUG1* is dispensable for basal endothelial function in relation to proliferation, apoptosis, barrier function, migration, mitochondrial function and inflammation.

## *TUG1* lncRNA is not required for basal sprouting, but relevant for VEGFA-stimulated sprouting *in vitro*

Advanced aging is often accompanied by a decline in angiogenesis resulting in increased cardiovascular morbidity and mortality [49, 50]. Therefore, endothelial cell sprouting was assessed in an *in vitro* angiogenesis assay. The loss of *TUG1* had no effect on human EC sprouting under basal conditions *in vitro* (Fig 3A). Conversely, *TUG1* knockdown slightly, but significantly, reduced cumulative sprout length after VEGFA-stimulation. Moreover, RNA sequencing following GapmeR-mediated silencing of HUVECs (GapmeR Control vs. GapmeR TUG1) resulted in only minor changes in gene expression (number of feature counts: 13 upregulated and 7 downregulated genes), despite a robust reduction in *TUG1* levels (Fig 3B). Some of the significantly regulated targets from the RNA-seq dataset were further analyzed by RT-qPCR (S2 Fig). None of these were robustly regulated by both GapmeRs. Hence, *TUG1* does not affect the transcriptional profile in ECs. This supported the results from the EC-specific functional assays, because *TUG1* silencing had no impact on the described characteristics.

Additionally, siRNAs were used to attenuate *TUG1* transcript levels in HUVECs. In more detail, two siRNAs resulted in a knockdown efficiency of more than 50% (siTUG1_1–37.62% ± 9.14%; siTUG1_2–45.25% ± 5.78%; Fig 3C). Basal sprouting was not changed after siRNA-mediated TUG1 knockdown, whereas VEGFA-stimulated sprout length was reduced by *TUG1* reduction (Fig 3D). Furthermore, TUG1-slienced HUVECs in combination with VEGFA stimulation did not result in any changes in migration compared to a VEGFA-stimulated control (S3 Fig).

In conclusion, *TUG1* is not relevant for the regulation of basal sprouting, but *TUG1* knockdown in combination with VEGFA stimulation decreased cumulative sprout length to a small extent as compared to VEGFA stimulation alone.

## TUG1 protein can be overexpressed in ECs, but is not involved in regulation of angiogenic sprouting

*TUG1* was recently described to encode a protein with a length of 153 amino acids [26, 27]. To see whether this predicted protein can be expressed in, and perhaps has a function in ECs, different constructs were generated for lentiviral overexpression (OE) in HUVECs. Three different inserts were subcloned from pEF1a plasmids [26] into the pLenti4-V5 backbone (Fig 4A).

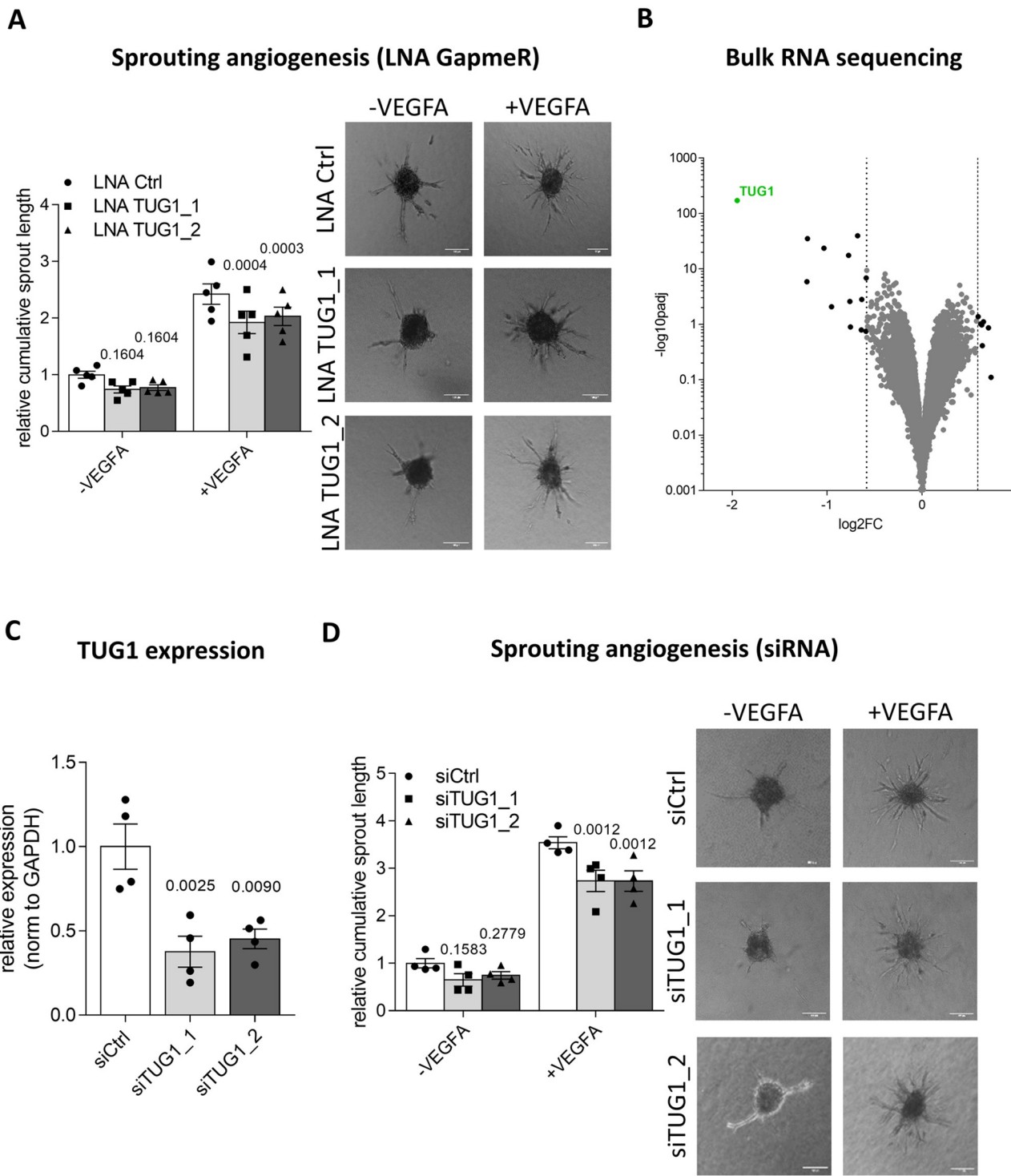

**Fig 3. *TUG1* influences VEGFA-stimulated sprouting. (A)** Quantification of cumulative sprout length by *in vitro* spheroid-assay after LNA GapmeR-mediated *TUG1* knockdown under basal conditions or with VEGFA stimulation (50 ng/ml for 24 h) in HUVECs. Representative images show the extent of sprouting as compared to 200 μm size bar (n = 5; SEM; RM two-way ANOVA with Geisser-Greenhouse correction and Holm-Sidak multiple comparison test). **(B)** Volcano plot of deregulated genes (log2 fold change vs.–log10 adjusted p-value) based on HUVEC bulk RNA-sequencing data (LNA Ctrl vs. LNA TUG1; n = 3 vs. 3). *TUG1* is represented by green dot. **(C)** *TUG1* expression levels in HUVECs 48h after siRNA transfection using RT-qPCR. Expression is relative to GAPDH (n = 4; SEM; RM One-way ANOVA with Holm-Sidak correction). **(D)** Quantification of cumulative sprout length by *in vitro* spheroid-assay after siRNA-mediated *TUG1* knockdown under basal conditions or with VEGFA stimulation (50 ng/ml for 24 h) in HUVECs. Representative images show the extent of sprouting as compared to 200 μm size bar (n = 4; SEM; RM two-way ANOVA with Geisser-Greenhouse correction and Holm-Sidak multiple comparison test).

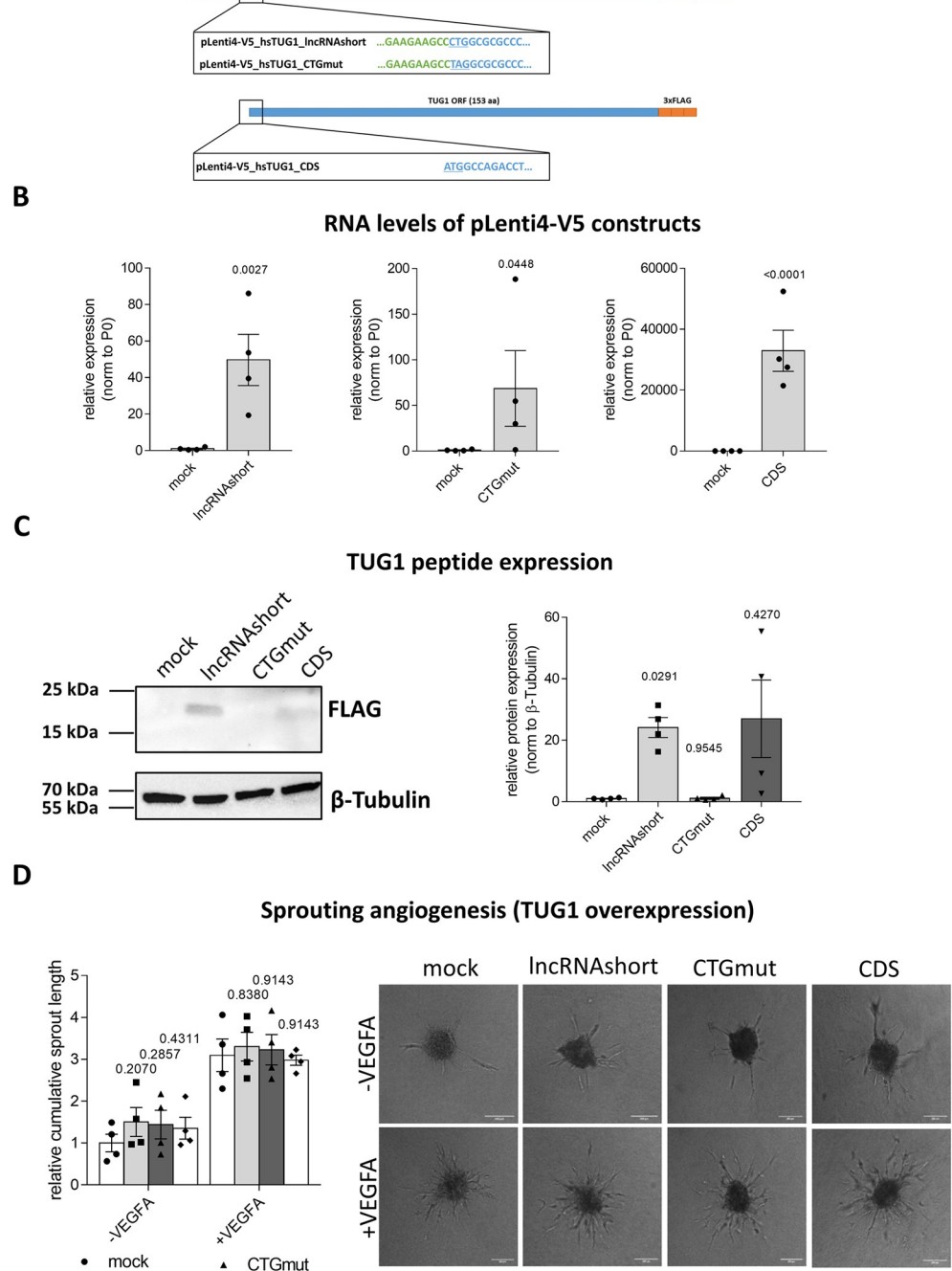

**Fig 4. TUG1 protein can be overexpressed in HUVECs, but is not involved in sprouting. (A)** Scheme of three pLenti4-V5 plasmids with different inserts for the *TUG1* lncRNA and protein coding open reading frame (ORF). PLenti4-V5_hsTUG1_lncRNAshort with the non-canonical start codon CTG representing the wild type sequence containing the information for the protein and the lncRNA, pLenti4-V5_hsTUG1_CTGmut with mutated start codon to stop codon (TAG) containing the information for the lncRNA only and pLenti4-V5_hsTUG1_CDS containing the codon optimized ORF for the TUG1 protein only. 5'- and 3'-untranslated region (UTR) indicated in green, ORF indicated in blue and C-terminal 3xFLAG-tag indicated in orange. Start of ORF sequence underlined and highlighted in blue. **(B)** RNA levels following lentiviral overexpression of the three different TUG1 proteins in HUVECs as determined by RT-qPCR (n = 4; SEM; ratio paired t-test). **(C)** Acquisition of TUG1 protein translation after exogenous

lentiviral overexpression of TUG1 proteins by Western Blot using anti-FLAG antibody (n = 4; SEM; RM one-way ANOVA with Geisser-Greenhouse correction and Holm-Sidak multiple comparison test). **(D)** Quantification of cumulative sprout length by *in vitro* spheroid-assay after lentiviral overexpression of TUG1 protein constructs under basal conditions or with VEGFA stimulation (50 ng/ml for 24 h) in HUVECs. Representative images show the extent of sprouting as compared to 200 μm size bar (n = 4; SEM; RM two-way ANOVA with Geisser-Greenhouse correction and Holm-Sidak multiple comparison test).

The construct named pLenti4-V5_hsTUG1_lncRNAshort contains almost the entire human *TUG1* lncRNA transcript, including parts of the endogenous 5'-UTR, the TUG1 protein open reading frame (ORF) with its non-canonical start codon (CTG) and a shortened 3'-UTR. The pLenti4-V5_hsTUG1_CTGmut plasmid differs from the former only by the replacement of the non-canonical start codon of the TUG1 ORF by TAG, which should prevent protein translation but leaves the remaining lncRNA intact. In contrast to the previous two, pLent4-V5_hsTUG1_CDS only contains the information for the TUG1 protein in form of the codon-optimized TUG1 ORF, but not for the lncRNA. In all plasmids a 3xFLAG-tag is inserted at the C-terminus of the TUG1 ORF resulting in a TUG1-3xFLAG-tag fusion protein in the case of translation.

Using these different constructs enables assigning certain effects to the lncRNA, the protein or both. After lentiviral transduction, RNA levels were expectedly strongly increased for all three constructs as determined by RT-qPCR (Fig 4B). Protein levels were assessed using anti-FLAG antibodies that target the C-terminal tag attached to the different proteins described in Fig 4A. As expected, the TUG1-3xFLAG-tag fusion protein was translated from the hsTUG1_lncRNAshort construct in HUVECs, whereas the CTG mutation in hsTUG1_CTGmut abolished TUG1 protein production (Fig 4C). The protein encoded by the hsTUG1_CDS construct was expressed to a smaller extent. Functionally, none of the constructs resulted in significant changes of sprout length in angiogenic sprouting assays *in vitro* compared to the control pLenti4-V5_mock (Fig 4D), neither under basal conditions nor after stimulation with VEGFA.

In summary, although the TUG1 protein could be translated following exogenous overexpression in HUVECs, it appeared not to be relevant for controlling angiogenesis. These results indicate that, regardless of *TUG1's* translation potential, *TUG1* is unlikely to regulate endothelial cell function *in vitro*.

## Discussion

This study identified that *TUG1* expression was attenuated by aging in human and mouse ECs. *TUG1* silencing had no effect on basal EC function including proliferation, apoptosis, barrier function, migration, mitochondrial function and monocyte adhesion, while VEGFA-stimulated sprouting was decreased significantly. Furthermore, *TUG1* did not influence the transcriptional profile. The TUG1 proteins (encoded by the hsTUG1_lncRNAshort and hsTUG1_CDS constructs) were translated in HUVECs following lentiviral overexpression, while overexpression of the construct with a mutated start codon (hsTUG1_CTGmut) did not result in detectable TUG1 protein. The TUG1 proteins did not regulate basal or VEGFA stimulated angiogenic sprouting *in vitro*.

*TUG1* is an interesting lncRNA because of a remarkable combination of features: *TUG1* was highly and ubiquitously expressed in multiple cell types and conserved among many different species [27]. Our results further showed an equal distribution in nucleus and cytoplasm (Fig 1E) and a regulation by aging in human (Fig 1C) and mouse (Fig 1D). Cardiovascular aging is accompanied by stiffening of the vessel wall, thickening of the intima, endothelial dysfunction and increased vascular inflammation [14]. This functional decline of ECs is caused by

oxidative stress, epigenetic changes, endothelial dysfunction and genomic instability [47]. We expected alterations in at least some of these characteristics following silencing of *TUG1*. Therefore, GapmeRs were used to target all *TUG1* transcripts (nuclear or cytoplasmic; Fig 2A) for the simulation of aged ECs. Unexpectedly, loss of *TUG1* did not change any phenotypic parameters related to aging in ECs under basal conditions (Fig 2B–2G). Instead, *TUG1* manipulation only resulted in a slight decrease of VEGFA-stimulated sprouting by using GapmeRs or siRNAs (Fig 3A and 3D). Manipulation of previously studied lncRNAs in loss-of-function studies resulted in stronger attenuation of angiogenic sprouting also at basal level [1, 51, 52]. Thus, the absence of effects on basal EC function after loss of *TUG1* represents a novelty. These findings were further supported: *TUG1* manipulation in combination with VEGFA treatment did not alter migration as assessed in an ECIS setup following lethal electroporation (S3 Fig). Consequently, the reduction in VEGFA-stimulated sprouting in TUG1-silenced HUVECs did not results from a decreased migratory capacity.

The dispensability of *TUG1* under basal conditions was further underlined by the results from bulk RNA-sequencing of control vs. *TUG1* knockdown. *TUG1* was the most robustly downregulated gene, while only very few differentially regulated genes resulted from the analysis of bulk RNA-sequencing data (S2 Fig). Consequently, *TUG1* did not contribute to transcriptional regulation in HUVECs under basal conditions. *In vivo* data showed that global *Tug1* knockout had no phenotype except for male infertility caused by impaired spermatogenesis with defects in number of sperms and abnormal sperm morphology [27]. Consequently, we do not expect a phenotype in angiogenesis.

Interestingly, the regulation by aging was the only significant upstream effect to be involved in the regulation of *TUG1* expression. Neither oxidized low density lipoprotein (oxLDL) nor hydrogen peroxide ($H_2O_2$)–both oxidative stressors–influenced *TUG1* RNA levels (S1A and S1B Fig). Furthermore, activating stimuli such as VEGFA and TNF-α or mechanic forces represented by shear stress did not change *TUG1* expression in HUVECs (S1C–S1E Fig). *TUG1* was previously described to be induced by Notch1 which is accompanied by promotion of self-renewal of glioma stem cells [21]. *Delta Like Canonical Notch Ligand 4* (*Dll4*)–which is an established activator of the Notch pathway in endothelial cells–was not able to induce *TUG1* expression in HUVECs (S1F Fig).

Even though we did not find evidence for a role of *TUG1* in ECs under normal culture conditions, *TUG1* might play a role under certain stress stimuli. The high levels of *TUG1* transcript might serve as a backup for certain stress responses. This was further supported by the findings of Dumbović et al. [46]: Intron retention in the *TUG1* transcript drives nuclear compartmentalization and the authors hypothesize that this might indeed serve as a system for buffering the *TUG1* transcript in particular stress conditions. In addition, *TUG1* is involved in diabetic retinopathy in mice [6] and many types of cancer via a nuclear or cytoplasmic function [21, 30, 53–55]. Taken together, these findings hint towards a cell- or context-specific function of *TUG1*.

Recently, translation of a TUG1 protein was revealed by which the TUG1 gene might exert an additional function [26]. Three different lentiviral constructs were generated depicting the *TUG1* transcript (hsTUG1_CTGmut), the TUG1 protein (hsTUG1_CDS) or both (hsTUG1_lncRNAshort) with a C-terminal FLAG-tag. None of the overexpressed constructs changed angiogenic sprouting *in vitro* under basal or VEGFA-stimulated conditions significantly. According to Lewandowski et al. [27], the *trans*-based function of the TUG1 lncRNA is negligible, whereas the TUG1 protein is involved in the regulation of mitochondrial bioenergetics. We could not identify a role of *TUG1* in mitochondrial function in ECs (Fig 2F).

A limitation of the study addresses the loss-of-function studies via transfection using siRNAs or gapmeRs being only transient instead of applying the CRISPR/Cas system. The same

affects the lentiviral overexpression of the TUG1 proteins which was stable short-term. The reason, why we used these methods for knockdown or overexpression results from the fact that HUVECs are primary cells and can only be grown for several passages unlike immortalized cell lines. Thus, considering time for transduction and selection, HUVECs will have stopped to proliferate before or during the actual experiment.

Furthermore, the assays addressing EC function were performed exclusively *in vitro*. We do not expect any effects of *TUG1* manipulation in ECs *in vivo* which is in concordance with the phenotype in ECs in global TUG1$^{-/-}$ mice [27]. Consequently, the experiments that we performed *in vitro* can be transferred to and reproduced *in vivo*.

The glucose levels (5.7 mM) in the HUVEC medium (EBM, Lonza) did not reach the high glucose levels of 25 mM from [6]. Under these low glucose levels (considered as basal) HUVECs do not show deviation in regard to any EC specific function. However, this study does not address the effect of high glucose levels in combination with *TUG1* silencing on EC function.

In summary, we show that despite a high abundance and conservation of the lncRNA *TUG1* and the encoded protein, both are not essential for basal EC function. The small, but significant, contribution of the *TUG1* lncRNA to VEGF-induced endothelial cell sprouting, likely only influences endothelial cell function to a minor extent, if any.

## Supporting information

**S1 Fig. TUG1 is not regulated by EC activation, induction of quiescence, oxidative stress or inflammation.** TUG1 RNA levels were measured by RT-qPCR after stimulation with **(A)** 50 µg/ml oxLDL for 48h (n = 3; SEM; unpaired t-test), **(B)** 200 µM $H_2O_2$ for 1h (n = 3; SEM; unpaired t-test), **(C)** 50 ng/ml VEGFA for 24h (n = 4; SEM; paired t-test), **(D)** TNFa 10 ng/ml for 24h (n = 3–4; SEM; unpaired t-test), **(E)** shear stress with 20 Dyn/cm$^2$ for 72h (n = 8; SEM; paired t-test; cells treated for the same time under static conditions were taken along as Ctrl) and **(F)** 1 µg/ml rDll4 for 24h (n = 3; SEM; unpaired t-test; Hes Family BHLH Transcription Factor 1 (HES1) served as a Ctrl). Expression is normalized to GAPDH or TBP as determined by RT-qPCR.
(TIF)

**S2 Fig. RT-qPCR-based confirmation of RNA-sequencing results using both LNA GapmeRs against TUG1.** HUVECs were transfected with two LNA GapmeRs against TUG1—LNA TUG1_1 and LNA TUG1_2 –and LNA Ctrl and expression levels of **(A)** VAMP4, **(B)** TOR1AIP2, **(C)** KAT6B and **(D)** ABCA1 were measured after 48 hours by RT-qPCR. Expression is relative to P0 (n = 4; SEM; RM one-way ANOVA with Greenhouse-Geisser correction and Holm-Sidak multiple comparison test).
(TIF)

**S3 Fig. TUG1 does not influence migration in combination with VEGFA treatment.** A confluent monolayer of transfected HUVECs (LNA TUG1_1, LNA TUG1_2 or LNA Ctrl) were wounded in an ECIS setup and reestablishment was analyzed (n = 3; SEM; RM one-way ANOVA with Greenhouse-Geisser correction and Holm-Sidak multiple comparison test).
(TIF)

**S1 Raw image. Uncropped western blots.** Uncropped images of the western blots used in Fig 4.
(PDF)

**S1 Table. Details of reagents.** Oligonucleotide sequences used for RT-qPCR, as well as for cloning and sequencing are listed. Sequences that were used to synthesize siRNAs and LNA

gapmers can be found here. There is also a list of the antibodies and their dilutions used to generate western blots.
(TIF)

**S1 Data.**
(XLSX)

## Author Contributions

**Conceptualization:** Anna Theresa Gimbel, Kosta Theodorou, Norbert Hübner, Lars Maegdefessel, Stefanie Dimmeler, Sebastiaan van Heesch, Reinier A. Boon.

**Data curation:** Anna Theresa Gimbel.

**Formal analysis:** Anna Theresa Gimbel, Laura Stanicek, Veerle Kremer, Tamer Ali, Stefan Günther, Sandeep Kumar, Hanjoong Jo, Reinier A. Boon.

**Funding acquisition:** Norbert Hübner, Reinier A. Boon.

**Investigation:** Susanne Koziarek, Kosta Theodorou, Veerle Kremer, Sebastiaan van Heesch.

**Methodology:** Anna Theresa Gimbel, Susanne Koziarek, Jana Felicitas Schulz, Laura Stanicek, Tamer Ali, Stefan Günther, Sandeep Kumar, Hanjoong Jo, Norbert Hübner, Lars Maegdefessel, Sebastiaan van Heesch, Reinier A. Boon.

**Project administration:** Reinier A. Boon.

**Supervision:** Stefanie Dimmeler, Reinier A. Boon.

**Visualization:** Anna Theresa Gimbel, Jana Felicitas Schulz, Reinier A. Boon.

**Writing – original draft:** Anna Theresa Gimbel, Sebastiaan van Heesch, Reinier A. Boon.

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
