## [Decision Letter · Decision Letter 0]

30 Mar 2022

PONE-D-22-05524Aging-regulated TUG1 is dispensable for endothelial cell functionPLOS ONE

Dear Dr. Boon,

Thank you for submitting your manuscript to PLOS ONE. After careful consideration, we feel that it has merit but does not fully meet PLOS ONE’s publication criteria as it currently stands. Therefore, we invite you to submit a revised version of the manuscript that addresses the points raised during the review process.

We look forward to receiving your revised manuscript.

Kind regards,

Kishore K Wary, PhD

Academic Editor

PLOS ONE

Journal Requirements:

[No]. 

5. Please upload a copy of Supporting Information Figure S2 which you refer to in your text on page 27.

Additional Editor Comments:

Please address reviewers questions and concerns clearly and adequately.

Reviewers' comments:

Reviewer's Responses to Questions

**Comments to the Author**

1. Is the manuscript technically sound, and do the data support the conclusions?

Reviewer #1: Yes

Reviewer #2: Yes

2. Has the statistical analysis been performed appropriately and rigorously? 

Reviewer #1: Yes

Reviewer #2: Yes

3. Have the authors made all data underlying the findings in their manuscript fully available?

Reviewer #1: Yes

Reviewer #2: No

4. Is the manuscript presented in an intelligible fashion and written in standard English?

Reviewer #1: Yes

Reviewer #2: Yes

5. Review Comments to the Author

Reviewer #1: Gimbel et al. examined the role of TUG1 long non-coding RNA in endothelial functions. TUG1 is well preserved between rodents and humans, and its expression is reduced during organisms’ aging as well as cell passaging.

The authors conducted knockdown and overexpression in HUVEC and found that TUG1 is not involved in proliferation, apoptosis, migration, barrier function, mitochondrial function, and inflammation. VEGF-induced sprouting angiogenesis is reduced by TUG1 knockdown. It is consistent in two knockdown protocols. However, TUG overexpression did not increase VEGF-induced sprouting.

The research group sought upstream regulator for TUG1 expression. Oxidative stress did not influence. VEGF-A and TNF did not change TUG1, either. Notch activator did not change TUG1, either.

The authors concluded cell- and context-specific function of TUG1 because of its significant functions in other cell types like podocytes of kidney and cancer cells.

Well written methodology and results. Interpretation of the data is reasonable. The conclusion is supported by the data. I would suggest further discussion of the study limitation in comparison with previous studies demonstrated positive results.

The functional studies in this study rely on the short-term transduction —24 h transfection and 3 day maintenance. If we compare to the method in the paper (ref. 6), the use of short-term transduction may be a limitation. In this paper, stable clones were generated using CRISPR/Cas9 system including knockout and overexpression. Also, in vivo overexpression was performed to demonstrate the overexpression rescue TUG1-dependent phenotype in vivo diabetic milieu of db/db mice. Importance of in vivo model is suggested by the negative results that in vitro stimulation and senescence doesn’t reproduce TUG1 reduction, which is shown in vivo aging condition.

Another limitation may be lack of the use of high glucose condition. PGC-1 and mitochondria biogenesis is downstream target of TUG1 by ref. 6. Although knockdown in normal glucose showed significant changes in podocytes in this paper. High glucose culture (25 mM glucose) was used to test the effect of TUG1 overexpression. This suggests that TUG1 may be significant in metabolically challenged conditions in high glucose, acidic environment and hypoxia.

By the way, glucose concentration is not disclosed. It should be written.

Reviewer #2: Summary: The current research article by Gimbel et al presents an interesting account on the role of Taurine upregulated Gene 1 (Tug1) in Endothelial cells (ECs) in context to aging-induced cardiovascular diseases. They demonstrated peculiar expression pattern of Tug1 aging endothelial cells compared to young mouse and/or early passage of human ECs. However, the silencing of Tug1 has no effect on the homeostatic EC function. This is very interesting that, the attenuation of Tug1 in aged human and mouse ECs is dispensable for EC function.

Strengths of the work: Projected evidence in favor of Tug1 expression in young vs aged mouse and human ECs (early vs late passage). Secondly the authors are providing the strong data in terms of homeostatic function of Tug1 in ECs by studying various functional parameters like proliferation, apoptosis, barrier function, migration, mitochondrial function, and inflammatory function. Authors present strong evidence in relation to the contribution of Tug1 with VEGF challenge in inducing endothelial cell sprouting, the effect is minimal, though it is significant.

Overall: In my opinion, the manuscript is interesting, and the experiments appear carefully conducted using both in vivo and in vitro approach. I have few concerns.

• Does VEGF challenge decreases EC migration along with sprouting in Tug1 silenced ECs? If so, how this is relevant to aging-induced CVD, discuss.

• What are the differential feature counts of LNA ctrl vs LNA TUG1 from bulk RNA seq data?

• There is no mention of data availability in repository.

6. PLOS authors have the option to publish the peer review history of their article (what does this mean?). If published, this will include your full peer review and any attached files.

Reviewer #1: No

Reviewer #2: No

---

## [Author Response · Author response to Decision Letter 0]

21 Jun 2022

Rebuttal Letter

Journal Requirements: 

We adjusted the formatting according to the PLOS ONE guidelines. 

[No]. 

We changed the section “Competing Interests with the statement: “The authors have declared that no competing interest exist.”

An Excel sheet with raw data and analysis is attached as supplementary data file.

We uploaded the uncropped and unadjusted images from Western Blots as “S1_raw_images.tif”. 

4. Please upload a copy of Supporting Information Figure S2 which you refer to in your text on page 27.

We attached the Figure “S2 Fig” as a .tif file.

We are not aware of using retracted references.

Additional Editor Comments:

Please address reviewers’ questions and concerns clearly and adequately.

Reviewer's Responses to Questions 

Comments to the Author

1. Is the manuscript technically sound, and do the data support the conclusions?

Reviewer #1: Yes

Reviewer #2: Yes

2. Has the statistical analysis been performed appropriately and rigorously? 

Reviewer #1: Yes

Reviewer #2: Yes

3. Have the authors made all data underlying the findings in their manuscript fully available?

Reviewer #1: Yes

Reviewer #2: No

4. Is the manuscript presented in an intelligible fashion and written in standard English?

Reviewer #1: Yes

Reviewer #2: Yes

5. Review Comments to the Author

Reviewer #1: Gimbel et al. examined the role of TUG1 long non-coding RNA in endothelial functions. TUG1 is well preserved between rodents and humans, and its expression is reduced during organisms’ aging as well as cell passaging.

The authors conducted knockdown and overexpression in HUVEC and found that TUG1 is not involved in proliferation, apoptosis, migration, barrier function, mitochondrial function, and inflammation. VEGF-induced sprouting angiogenesis is reduced by TUG1 knockdown. It is consistent in two knockdown protocols. However, TUG overexpression did not increase VEGF-induced sprouting.

The research group sought upstream regulator for TUG1 expression. Oxidative stress did not influence. VEGF-A and TNF did not change TUG1, either. Notch activator did not change TUG1, either.

The authors concluded cell- and context-specific function of TUG1 because of its significant functions in other cell types like podocytes of kidney and cancer cells.

Well written methodology and results. Interpretation of the data is reasonable. The conclusion is supported by the data. I would suggest further discussion of the study limitation in comparison with previous studies demonstrated positive results.

The functional studies in this study rely on the short-term transduction —24 h transfection and 3 day maintenance. If we compare to the method in the paper (ref. 6), the use of short-term transduction may be a limitation. In this paper, stable clones were generated using CRISPR/Cas9 system including knockout and overexpression. Also, in vivo overexpression was performed to demonstrate the overexpression rescue TUG1-dependent phenotype in vivo diabetic milieu of db/db mice. Importance of in vivo model is suggested by the negative results that in vitro stimulation and senescence doesn’t reproduce TUG1 reduction, which is shown in vivo aging condition.

Another limitation may be lack of the use of high glucose condition. PGC-1 and mitochondria biogenesis is downstream target of TUG1 by ref. 6. Although knockdown in normal glucose showed significant changes in podocytes in this paper. High glucose culture (25 mM glucose) was used to test the effect of TUG1 overexpression. This suggests that TUG1 may be significant in metabolically challenged conditions in high glucose, acidic environment and hypoxia.

By the way, glucose concentration is not disclosed. It should be written.

-We thank the reviewer for the suggestions to improve our manuscript. We adapted the discussion session and now included a paragraph on the limitations of the study based on the comments from reviewer #1:

We discussed the use of short-term transfection in HUVECs and its benefits compared to CRISPR/Cas system. 

We also discussed the reason for usage of assays only in vitro based on recently published data on TUG1-/- mice lacking any phenotype under basal conditions except for sperm deformation in male mice. 

The glucose concentration of the EBM medium was mentioned in the Discussion part and brought into context (page 30).

Reviewer #2: Summary: The current research article by Gimbel et al presents an interesting account on the role of Taurine upregulated Gene 1 (Tug1) in Endothelial cells (ECs) in context to aging-induced cardiovascular diseases. They demonstrated peculiar expression pattern of Tug1 aging endothelial cells compared to young mouse and/or early passage of human ECs. However, the silencing of Tug1 has no effect on the homeostatic EC function. This is very interesting that, the attenuation of Tug1 in aged human and mouse ECs is dispensable for EC function.

Strengths of the work: Projected evidence in favor of Tug1 expression in young vs aged mouse and human ECs (early vs late passage). Secondly the authors are providing the strong data in terms of homeostatic function of Tug1 in ECs by studying various functional parameters like proliferation, apoptosis, barrier function, migration, mitochondrial function, and inflammatory function. Authors present strong evidence in relation to the contribution of Tug1 with VEGF challenge in inducing endothelial cell sprouting, the effect is minimal, though it is significant.

Overall: In my opinion, the manuscript is interesting, and the experiments appear carefully conducted using both in vivo and in vitro approach. I have few concerns.

• Does VEGF challenge decreases EC migration along with sprouting in Tug1 silenced ECs? If so, how this is relevant to aging-induced CVD, discuss.

• What are the differential feature counts of LNA ctrl vs LNA TUG1 from bulk RNA seq data?

• There is no mention of data availability in repository.

-We thank the reviewer for these insightful suggestions for improving the manuscript.

We performed the ECIS experiment combining the effects of LNA GapmeR-mediated TUG1 knockdown and VEGFA stimulation. We could not detect any changes comparing Ctrl vs. TUG1 knockdown conditions (discussed on page 28; S3 Fig). 

We added the differential feature counts in the manuscript on page 22.

We added the raw data in the Supplementary information.

6. PLOS authors have the option to publish the peer review history of their article (what does this mean?). If published, this will include your full peer review and any attached files.

Do you want your identity to be public for this peer review? For information about this choice, including consent withdrawal, please see our Privacy Policy.

Reviewer #1: No

Reviewer #2: No

---

## [Editor Report · Decision Letter 1]

13 Jul 2022

Aging-regulated TUG1 is dispensable for endothelial cell function

PONE-D-22-05524R1

Dear Dr. %Boon%,

We’re pleased to inform you that your manuscript has been judged scientifically suitable for publication and will be formally accepted for publication once it meets all outstanding technical requirements.

Kind regards,

Kishore K Wary, PhD

Academic Editor

PLOS ONE

Additional Editor Comments (optional):

Through the revised manuscript is improved and the major comments have been addressed adequately.

Reviewers' comments:

None

---

## [Editor Report · Acceptance letter]

25 Aug 2022

PONE-D-22-05524R1 

Aging-regulated TUG1 is dispensable for endothelial cell function 

Dear Dr. Boon:

I'm pleased to inform you that your manuscript has been deemed suitable for publication in PLOS ONE. Congratulations! Your manuscript is now with our production department. 

Kind regards, 

on behalf of

Dr Kishore K Wary 

Academic Editor

PLOS ONE